# First Report of *Streptococcus agalactiae* Meningitis in a Non-Pregnant Adult in Italy

**DOI:** 10.3390/microorganisms13050978

**Published:** 2025-04-24

**Authors:** Giorgia Borriello, Giovanna Fusco, Francesca Greco, Maria Vittoria Mauro, Lorella Barca, Antonio Limone, Maria Garzi Cosentino, Agata Campione, Antonio Rinaldi, Saveria Dodaro, Esterina De Carlo, Sonia Greco, Valeria Vangeli, Rubina Paradiso, Antonio Mastroianni

**Affiliations:** 1Experimental Zooprophylactic Institute of Southern Italy, 80055 Portici, Italy; giorgia.borriello@izsmportici.it (G.B.); giovanna.fusco@izsmportici.it (G.F.); lorella.barca@izsmportici.it (L.B.); antolim@izsmportici.it (A.L.); maria.garzicosentino@izsmportici.it (M.G.C.); agata.campione@izsmportici.it (A.C.); antonio.rinaldi@izsmportici.it (A.R.); esterina.decarlo@izsmportici.it (E.D.C.); 2Microbiology & Virology Unit, “Annunziata” Hub Hospital, 87100 Cosenza, Italy; francesca.greco@aocs.it (F.G.); m.mauro@aocs.it (M.V.M.); s.dodaro@aocs.it (S.D.); 3Infectious & Tropical Diseases Unit, “Annunziata” Hub Hospital, 87100 Cosenza, Italy; grecosonia1976@gmail.com (S.G.); valeriavangeli@gmail.com (V.V.); antoniomastroianni@yahoo.it (A.M.)

**Keywords:** *Streptococcus agalactiae*, meningitis, risk factors, pathogenicity, whole genome sequencing

## Abstract

This study, for the first time in Italy, analyses by WGS a *Streptococcus agalactiae* strain isolated from a non-pregnant adult affected by Meningitis and without common risk factors. The *S. agalactiae* strain was classified as a serotype II (SS2), sequence type ST569. Molecular characterization evidenced the presence of resistance genes to tetracycline and macrolide (*tet*(M) and *mre*(A)) and several virulence genes coding for adhesion and immune evasion factors (*bca*, *cps* family, *neu* family, *scpB*, *gbs* family, *pil* family and *hylB*), toxins (cfa/*cfb*, *cyl* family), pro-inflammatory factors (*lepA*), and two homologous genes that contributed to bacterial escape from the host immune system (*lmb*, *luxS*). SNP analysis showed 18 different alleles, with 9 missense SNP mutations related to genes involved in cellular metabolism (*dhaS*, *ftsE*, *ligA*, *nrdD* and *secA*), virulence (*bgrR* and *galE*) and antimicrobial resistance (*glpK* and *mutL*). SNPs in *glpK* and *mutL* genes might reduce susceptibility to drugs. The SNP analysis highlighted the presence of mutations conferring pathogenicity to the strain. The evidence in this study could explain the development of Meningitis in a healthy patient. This case highlights the importance of using molecular methods to characterize the complete genome of a bacterial species that could seriously affect human health.

## 1. Introduction

*Streptococcus agalactiae* is a Gram-positive, coccus-shaped, catalase-negative and oxidase-negative microorganism. This pathogen is also commonly known as group B Streptococcus (GBS) since it belongs to Lancefield’s group B classification, which includes streptococci that harbour a specific polysaccharide in their bacterial wall [1,2,3]. *S. agalactiae* was first isolated in 1887 by Edmond Nocard from the milk of mastitis-affected cattle. However, its zoonotic potential and ability to cause severe disease in humans was only discovered in 1934, when Fry isolated it from three human cases who died from postpartum sepsis [4,5]. *S. agalactiae* lives as a commensal bacterium in the genitourinary and gastrointestinal systems of humans and can cause serious disease, especially in immunosuppressed subjects (i.e., elderly and children) [4]. Children are most affected by GBS, and the onset of illness is characterized by sepsis, pneumonia and Meningitis [6]. In 1973, an intrapartum prophylaxis program was initiated to limit cases of illness and death in newborns by administering antibiotics during pregnancy [7]. Nevertheless, it is estimated that GBS is the cause worldwide of 1% of all stillborn babies in developed countries and 4% of all stillborn babies in African countries [8]. It is well-known that GBS is also a pathogen that infects different animal species, such as cattle, dogs, cats, fish, dolphins, and crocodiles [9]. In cattle, *S. agalactiae* is one of the main responsible pathogens for mastitis in farmed animals worldwide [10]; several authors also suggest that the disease in cattle may occur via reverse zoonosis events [7,11]. The main virulence factors of GBS are the polysaccharides of the capsule and the beta-hemolytic toxin [2,6]. Depending on the immunological reactivity of the antigens of the capsule, 10 serotypes have been identified; the most prevalent in America and Europe are Ia, Ib, II, III, and V [12,13]. To date, 65% of GBS human cases of non-pregnant adults reported in Italy have concerned endocardial infections, resulting in embolism events and consequent neurological problems [14]. Presently, cases of GBS-associated Meningitis have never been described in Italy, and the case described in this paper is the first. However, in the literature, serotype V GBS-sustained infections in non-pregnant adults have been documented worldwide, and they usually include not only cases of Meningitis but also skin infections, osteomyelitis and endocarditis [15,16,17]. It should be noted that although GBS is responsible for a small portion of patients with acute bacterial Meningitis, 7–8% of non-pregnant adults with GBS bacteremia have Meningitis [18,19]. In the present study, the authors isolated a Group B *S. agalactiae* strain in a case of Meningitis in a 37-year-old patient with no previous health problems or common risk factors of infection, only a neurinoma of the left auditory nerve. The GBS strain was isolated from cerebrospinal fluid (CSF) and characterized by whole genome sequencing (WGS). The authors also describe a second case of GBS-associated Meningitis in an obese 50-year-old woman. However, the latter was detected only by Real-Time PCR as culture isolation from CSF failed.

## 2. Materials and Methods

### 2.1. Case Review

The first case—Human CSF: in October 2023, a male patient of approximately 37 years of age was admitted at the ‘Presidio Ospedaliero Annunziata’ hospital of Cosenza in the Calabria region of southern Italy due to symptoms referable to Meningitis (hereafter mentioned as patient one). The patient did not report previous health problems and was not affected by common risk factors of infection, such as type 2 diabetes mellitus, neoplasms, immunodeficiency and kidney disease [14,17]. The patient exhibited only a benign neurinoma of the left auditory nerve. The CSF was analyzed for (a) the characterization of biomedical values using the SYSMEX XN-550 (Sysmex Corporation, Kobe, Japan) instrument following the manufacturer’s instructions and for (b) the meningoencephalitis panel using the FilmArray Meningitis/Encephalitis panel (BioFire/bioMérieux, Salt Lake City, UT, USA) following the manufacturer’s instructions. The latter gave positive results. Microscopic examination revealed the presence of Gram-positive cocci. The patient was given antibiotic therapy as follows: vancomycin 1 g every 12 h plus clindamycin 900 mg every 8 h for 10 days.

Second case—Human CSF: in December 2023, a female obese patient of approximately 50 years of age was admitted at the ‘Presidio Ospedaliero Annunziata’ hospital of Cosenza in a confusional state (hereafter mentioned as patient two). Similarly, as in the first case, the CSF was analyzed for (a) the characterization of biochemical values using the SYSMEX XN-550 instrument and for (b) the meningoencephalitis panel using the FILMARRAYTM Meningitis/Enc kit (Biomerieux). The latter gave positive results. Microscopic examination revealed the presence of Gram-positive cocci. The patient received therapy with ceftriaxone 2 g every 12 h and vancomycin 1 g every 12 h for 10 days. (Table 1)

The samples used in the present study were collected during routine activities; therefore, according to the applicable regulations, neither ethical approval nor informed consent was required.

### 2.2. Bacteriological Examination

The two samples of CSF were inoculated into blood agar, chocolate agar, sabouraud agar, and thioglycolate. Subsequently, they were incubated at 37 °C for 24–48 h at 5% CO_2_. *S. agalactiae* was isolated and identified only from the CSF withdrawn from patient one: 24 h after incubation, beta haemolytic colonies were detected on blood agar plates and then identified as *S. agalactiae* by mass spectrometry (MALDI-TOF). GBS was not isolated from CSF withdrawn from patient two. A pure colony of *S. agalactiae* isolated from patient one and a sample of the CSF from patient two were sent to the IZSM for molecular characterisation.

### 2.3. DNA Extraction

Approximately 2 mL of CSF from patient two was centrifuged at 12,000 rpm for 10 min at room temperature. Then, the supernatant was discarded, and the pellet was subsequently used for DNA extraction. Typical colonies of GBS isolated from patient one were resuspended in 200 µL of PBS and then used for DNA extraction. DNA extraction from both samples was carried out using the DNeasy PowerSoil kit (Qiagen, Hilden, Germany) according to the manufacturer’s instructions and subsequently quantified using the Qubit fluorometer (Thermo Fisher Scientific, Waltham, MA, United States).

### 2.4. PCR Typing

A specific real-time PCR was carried out to assess the presence of *S. agalactiae* DNA in the CSF sample from patient two [20]. The PCR protocol included a forward primer (5′-GGGAACAGATTATGAAAAACCG-3′), a reverse primer (5′-AAGGCTTCTACACGACTACCAA-3′) and a probe (HEX-5′-AGACTTCATTGCGTGCCAACCCTGAGAC-3′-BHQ1) targeting the *cfb* gene. The reaction mixture was prepared in a final volume of 25 µL, including 1 µM of each primer, 0.5 µM of the probe, TaqMan Universal PCR Master Mix 1X (Applied Biosystems, Waltham, MA, USA) and 5 µL of DNA template. Thermocycling conditions consisted of an initial denaturation step at 95 °C for 10 min, followed by 45 cycles at 94 °C for 30 s and 60 °C for 1 min. Real-time PCR was carried out on a CFX 96 instrument (Bio-Rad Laboratories Inc., Hercules, CA, USA).

### 2.5. Whole Genome Sequencing Analysis

A DNA library was generated from the extracted DNA of the *S. agalactiae* strain isolated from patient one. The library was prepared using 150 ng of DNA with the Ion Xpress Fragment Library kit (Life Technologies, Carlsbad, CA, USA) following the manufacturer’s instructions and loaded on a sequencing chip by using the Ion Chef instrument (Thermo Fisher Scientific). The bacterial genome was sequenced at approximately 25× coverage. Sequencing was performed on the Ion Gene Studio S5 platform (Thermo Fisher Scientific) as described by the manufacturer’s protocol by generating 400 bp single-end reads. Reads were taxonomically classified using the Kraken 2 software v. 2.2.0 [21], and the RAST tool v. 1.3.0 [22] was used for genomic annotation. MLST v. 2.0.9 database (https://pubmlst.org/sagalactiae/; accessed on 12 April 2024) and the serotype bioinformatic tool v. 2.0.1 (https://github.com/aquacen/serotype_Sagalactiae; accessed on 12 April 2024) [23] were then used to assign ST and serotype, respectively. Genes encoding for virulence factors were detected by using VFDB (the virulence factor database, http://www.mgc.ac.cn/VFs/; accessed on 18 April 2024).

The core genome multilocus sequence typing (cgMLST) analysis was performed by chewbbaca (v. 3.3.4) using the pubmlst scheme (h_*S. agalactiae* v1.0) consisting of 1405 loci. Publicly available sequences of *S. agalactiae* serotype II strains (1056 isolates, Appendix A) were selected from human, animal and environmental samples collected worldwide and used as a database. The combination of all alleles in each strain was used to generate minimum spanning trees (MST) using the software GrapeTree (version 2.2) to identify related strains. Isolates assigned as sequence type ST569 were used for the SNP analysis and to investigate phylogenetic relationship. Finally, the Resfinder tool v. 4.6.0 was used to identify acquired genes (http://genepi.food.dtu.dk/resfinder; accessed on 18 April 2024).

## 3. Results

### 3.1. Biochemical Analysis of CSF Samples

Biochemical values of patient one: 6165 white blood cells/mm^3^ (90% neutrophils, 10% lymphocytes), IgG = 33.80 mg/dL (normal range: 0.00–3.40 mg/dL), albumin = 219 mg/dL (normal range: 0.00–35 mg/dL), glucose = 53 mg/dL (normal range: 40–70 mg/dL).

Biochemical values of patient two: 6980 white blood cells/mm^3^ (88.1% neutrophils, 11.9% lymphocytes), IgG = 100 mg/dL (normal range: 0.00–3.40 mg/dL), albumin = 483 mg/dL (normal range: 0.00–35 mg/dL), glucose = 69 mg/dL (normal range: 40–70 mg/dL).

### 3.2. Microbiological Examination and Molecular Analysis of CSF Samples

Only one of the two CSF samples under study resulted positive for bacteriological examination; the *S. agalactiae* strain was isolated from patient one and confirmed by mass spectrometry. Nevertheless, both CSF samples were tested using a real-time PCR analysis targeting the cfb gene, and the results were positive.

### 3.3. Whole Genome Sequencing Analysis

The *S. agalactiae* strain isolated from patient one was processed for WGS characterization by high-throughput sequencing. The WGS analysis produced a total of 280,543,897 bp, distributed among 983,026 reads. The assembly resulted in 69 contigs, accounting for 2,002,932 bp and an N50 of 111,704 bp. The genome annotation exhibited 2174 protein-coding sequences (CDS), 52 transfer RNA (tRNA) genes, and six ribosomal RNA (rRNA) genes. PATRIC annotation included 313 hypothetical proteins and 1861 proteins with functional assignments. The functional assignments included 614 proteins with Enzyme Commission (EC) numbers, 510 with Gene Ontology (GO) assignments, and 432 proteins mapped to KEGG pathways.

The MLST analysis based on a seven genes scheme (adhP, atr, glcK, glnA, pheS, sdhA, tkt) identified the analyzed genome as Sequence Type ST569, while in silico serotype typing, based on capsular polysaccharide (CPS) genes, assigned the strain to serotype II.

The cgMLST analysis grouped the isolates into a total of 29 different STs with a variable distribution within serotype II. The most frequent STs were ST1 (*n* = 31), ST2 (*n* = 37), ST10 (*n* = 69), ST12 (*n* = 58), ST22 (*n* = 301), ST28 (*n* = 210), ST61 (*n* = 53) and ST554 (*n* = 26). The ST569 cluster included our strain (S_25803) and three other isolates (Figure 1).

These three strains were of human origin and were isolated one from the USA and two from France: the American strain was isolated from an invasive disease in 2016, one of the French strains was isolated from a bacteremia case in 2009, and the second French strain was isolated in 2020 from a non-specified disease.

From the sequenced *S. agalactiae* genome, the presence of virulence genes and antimicrobial drug-resistance genes was investigated. Bacterial virulence factors confer on the harbouring strain the ability to invade and colonize the host, thus increasing its ability to induce infection. The strain under study displayed the presence of several virulence genes coding for adhesion and immune evasion factors (*bca*, *cps* family, *neu* family, *scpB*, *gbs* family, *pil* family and *hylB*), toxins (*cfa*/*cfb*, *cyl* family), pro-inflammatory factors (*lepA*), and two homologous virulence genes contributing to bacterial escape from the host immune system (*lmb*, *luxS*). Genetic characterization of AMR genes showed the presence of *mre*(A) and *tet*(M) genes responsible for genetic resistance to macrolides and tetracyclines, respectively. These antibiotic resistances were also confirmed by disk diffusion test and MIC determination (erythromycin MIC > 0.25 mcg/mL; tetracycline MIC > 8 mcg/mL). The analyzed genome was also used to perform a SNPs analysis using the GCA_014875295 strain as a reference. The results showed a total of 18 different alleles, with 9 missense SNP mutations related to genes involved in cellular metabolism (*dhaS*, *ftsE*, *ligA*, *nrdD* and *secA*), virulence (*bgrR* and *galE*) and antimicrobial resistance (*glpK* and *mutL*). SNPs in *glpK* and *mutL* genes might reduce susceptibility to drugs.

## 4. Discussion

This study describes the WGS-based characterization of a *Streptococcus agalactiae* strain isolated from the cerebrospinal fluid of a meningitis-affected 37-year-old man in southern Italy. The patient did not report previous health problems, nor did neither suffered from type II diabetes mellitus, malignant neoplasms, immunodeficiency or kidney disease. Based on CPS genes and MLST analysis, the strain under study was identified as serotype II and Sequence Type ST569, respectively. Phylogenetic analysis indicated that the strain could be clustered with three other strains, all isolated from human cases in the USA and France.

GBS is a commensal bacterium of the human microflora and frequently colonizes the gastrointestinal, respiratory, and vaginal tracts [24]. The maternal colonization of the gastrointestinal tract and/or lower reproductive tract is considered a primary risk factor for neonatal invasive diseases such as sepsis, Meningitis, and peripartum infection [25]. On the other hand, the incidence of invasive infections associated with GBS in non-pregnant adults is rising worldwide. Studies in the literature report a cumulative incidence rate of invasive GBS disease ranging from 2.4 to 9.2 cases per 100,000 non-pregnant adults [26,27]. This pathogen can induce a range of different invasive diseases, such as skin and osteoarticular infections, pneumonia, urosepsis, endocarditis, peritonitis, Meningitis, and streptococcal toxic shock syndrome (STSS) [28]. Specifically, GBS is responsible for 0.3–4.3% of the total cases of Meningitis in non-pregnant adults [27]. However, adults develop the disease less frequently compared to children because GBS does not easily pass the blood-brain barrier [16]. Among non-pregnant adults, the most affected categories are the elderly and immunosuppressed [14]. The mortality rate of non-pregnant adults belonging to risk categories may exceed 50% [29]. Primary risk factors for the development of GBS meningitis in non-pregnant adults include type II diabetes mellitus, neoplasms, immunodeficiency, and kidney disease [14,17]. In a study conducted on non-pregnant adults in Reunion Island, Camuset et al. found a higher incidence rate of GBS infections compared to elsewhere; obesity was one of the most important risk factors associated with the disease [30]. Similarly, an epidemiological study in the USA reported an increased incidence of invasive GBS infections in non-pregnant adults, ranging from 8.1% in 2008 to 10.9% in 2016, associated with obesity or diabetes [31]. In addition, a recent case of GBS meningitis in a non-pregnant patient with a history of hypertension, dyslipidemia, overweight, and alcohol consumption was described in Portugal [32]. In our study, patient two was hospitalized, reporting only a confusional state and no other predisposing conditions except for obesity. This, together with the existing literature mentioned above, suggests that obesity is an important risk factor for the development of GBS meningitis. However, it must be noted that in inpatient two, it was not possible to detect *S. agalactiae* with conventional culture methods, and its presence was confirmed only by Real-Time PCR. Hence, it can be assumed that *S. agalactiae* could have been present in both samples even though, in inpatient two, the bacteriological examination was negative. Failure of bacteriological isolation of the microorganism could be explained by the patient taking an antibiotic treatment before hospitalization, reducing the sensitivity of the cultural examination. This evidence highlights the importance of the use of real-time PCR molecular tests: they could be more sensitive in detecting Streptococcus than bacteriological tests when antibiotic treatment is suspected.

On the other hand, patient one was hospitalized, presenting fever, headache, and diarrhoea. However, they did not present any common risk factor for invasive GBS infection, except for a benign neurinoma of the left auditory nerve. The injury of the acoustic apparatus could have been a factor that permitted *S. agalactiae* to reach the brain, as already reported elsewhere [33]. Recently, in the USA, cases of GBS-associated Meningitis in non-pregnant adults affected by otitis have already been described [16,34].

The WGS characterization of the strain under study exhibited the presence of numerous virulence factors, which are essential to confer the bacteria the ability to induce invasive disease strategies and are responsible for its survival and colonization in the host [35]. The fact that patient one did not display any of the most common risk factors associated with GBS invasive disease highlights the important role these virulence factors have in determining the pathogenicity of the studied strain. Moreover, among the virulence factors harboured by the strain, SNP analysis identified the presence of missense mutations in the *bgrR* and galE genes, the former associated with increased virulence properties [36,37]. Consistently, cgMLST analysis indicated that the strain under study could be classified as ST569 (a rare Sequence Type) found only in three other cases, all grouped in the same cluster and responsible for infection and disease in humans [19]. However, it is important to note that these other strains were isolated in pregnant women and newborns [38]. The analysis of the genomes available in GenBank showed that ST569 is a unique sequence type in Italy; only one other *S. agalactiae* strain, serotype II, was identified but was classified as ST28, while other *S. agalactiae* strains resulted as serotypes III and IV.

Given the increasing incidence of GBS invasive infections in non-pregnant adults [30] and the zoonotic nature of the microorganism, it is recommended to manage and monitor the disease with a One Health approach when diagnosing both animal and human cases that can derive from both risk and non-risk categories.

## 5. Conclusions

*S. agalactiae* meningitis in non-pregnant adults is a rare disease, even though recent studies suggest it may become more prevalent, increasing the relevance of this case series. This study helps support the enhancement of diagnosis and treatment for Group B *S. agalactiae* meningitis and highlights the importance of considering GBS in all meningitis human cases for appropriate, timely diagnosis and treatment.

## Figures and Tables

**Figure 1 microorganisms-13-00978-f001:**
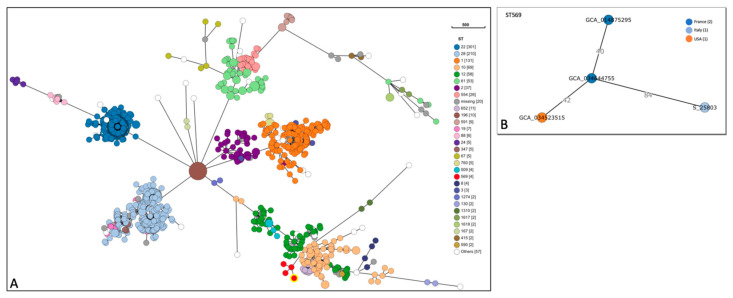
Minimum spanning of all *Streptococcus agalactiae* isolates. Phylogenetic tree based on the cgMLST data from publicly available *Streptococcus agalactiae* serotype II isolates (*n*  =  1056) of human, animal and environmental origin. Colours represent the different distributions of the sequence types. Each circle indicates an allele profile with lines connecting closely related isolates forming clusters (**A**). The ST569 (red circle) cluster was composed of four strains, including the one presented in this study. The lines indicate allelic differences (**B**).

**Table 1 microorganisms-13-00978-t001:** Summary of human cases with Meningitis infected by *Streptococcus agalactiae*. The table includes information on patient age, sex, immune status, pre-existing conditions, and microbiological results.

Summary of Human Cases with Meningitis Infected by GBS					
Case	Age	Sex	Immune Status	Pre-Existing Conditions	Microbiological Results
1	37 years old	Male	Normal	Benign neurinoma of the left acoustic nerve	Biochemical analysis with SYSMEX XN-550, Meningitis/Encephalitis panel FILMARRAY™ (Biomerieux). Positive for Gram-positive cocci
2	50 years old	Female	Possible immune compromise (obesity)	No pre-existing conditions reported	Biochemical analysis with SYSMEX XN-550, Meningitis/Encephalitis panel FILMARRAY™ (Biomerieux). Positive for Gram-positive cocci

## Data Availability

The data that support the findings of this study are openly available in *Streptococcus agalactiae* strain 25803HSAGAL chromosome at https://www.ncbi.nlm.nih.gov/nuccore/2731779437 (accessed on 12 May 2024), reference number CP154875.

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
