# Peer review of "First Report of Streptococcus agalactiae Meningitis in a Non-Pregnant Adult in Italy"

_microorganisms, 2025, doi:10.3390/microorganisms13050978_

Round 1
Reviewer 1 Report
Comments and Suggestions for Authors
The manuscript describes important data on the isolation, for the first time in Italy, of a strain of Streptococcus agalactiae isolated from a non-pregnant adult affected by meningitis and without common risk factors. Furthermore, the S. agalactiae strain was classified as serotype II, belonging to the rare sequence type ST569, evidencing the presence of tetracycline and macrolide resistance genes and several virulence genes encoding adhesion and immune evasion factors, toxins, pro-inflammatory factors and two homologous genes that contributed to bacterial escape from the host immune system.
However, the authors should present:
1- Approval by the local and/or European Ethics Committee.
2- Include the table of primers used in the analyses of virulence genes encoding:
- adhesion and immune evasion factors,
- toxins,
- pro-inflammatory factors,
- homologous genes that contributed to bacterial escape from the host immune system.
3- Include data on the antimicrobials used in the treatment of patients, especially for patient 2, whose bacteriological culture was negative.
Author Response
1- Approval by the local and/or European Ethics Committee.
REPLY: These cases were collected as part of routine clinical activities, and as such, ethical approval or informed consent was not required, following applicable regulations.
2- Include the table of primers used in the analyses of virulence genes encoding:
- adhesion and immune evasion factors,
- toxins,
- pro-inflammatory factors,
- homologous genes that contributed to bacterial escape from the host immune system.
REPLY: The analysis of virulence genes was performed by analysing sequencing data. Specifically, the assembled contigs were mapped using a virulence gene database with the VFDB tool (the Virulence Factor Database, http://www.mgc.ac.cn/VFs/), as described in line 130. PCR was not used for the detection of virulence genes.
3- Include data on the antimicrobials used in the treatment of patients, especially for patient 2, whose bacteriological culture was negative.
REPLY: The manuscript has been implemented with the addition of the therapy prescribed to the patients (lines 79-81 and lines 88-89).
Reviewer 2 Report
Comments and Suggestions for Authors
Regarding to manuscript microorganisms-3530832, entitled “First report of Streptococcus agalactiae meningitis in a non-pregnant adult in Italy”, authors did genomic sequence analysis and immune analysis of the patient with meningitis.
However, authors only presented the data plainly and did not reported and discussed the data from FBS and patient more thoughtfully.
An infection shall be involved in two did directions: GBS and human. This manuscript can be very decent findings as a review manuscript in the following recommendations.
- Human:
- authors shall summary the patients with meningitis infected by GBS from
immune points: normal immune patients vs immune compromised patients vs the
data.in a Table.
- GBS
Authors shall collected all GBS strains to cause the meningitis and then to present
and discuss the data in the following area to demonstrate the importance of these two GBS isolates.
- Serotype and ST types
- The invasion genes for host immune system.
- Virulence genes

Author Response
- Human: authors shall summary the patients with meningitis infected by GBS from immune points: normal immune patients vs immune compromised patients vs the data.in a Table.
REPLY: manuscript was implemented with Table 1.
- GBS: Authors shall collect all GBS strains to cause the meningitis and then to present and discuss the data in the following area to demonstrate the importance of these two GBS isolates.
- Serotype and ST types
- The invasion genes for host immune system.
- Virulence genes
REPLY: all the genomes of the GBS strains included in figure 1 were downloaded from GeneBank and are all serotype 2, with different STs (as reported in the figure legend). Unfortunately, the metadata associated with these strains do not include information about meningitis. In particular, out of the 1056 strains, reported data specify that: 62 strains caused bacteremia, 1 strain caused bacterial infection, 1 strain bloodstream infection, 10 strains were classified as carriers, 17 strains “colonization”, 1 strain caused infection, 1 strain inhalatory syndrome, 529 strains invasive disease, 2 strains from healthy humans and for the remaining 432 strains no information is reported about associated diseases. Regarding the two GBS strains isolated from healthy subjects, deposited as ST1 and ST2 respectively, only the geographical origin was reported in the GeneBank database (SriLanca and Kenia, respectively). This lack of data therefore does not allow the elaboration required and only analysis of STs was reported in the manuscript. This limitation also applies to the previous request, about the analysis related to the human/immune system point.